# Wild Italian *Prunus spinosa* L. Fruit Exerts In Vitro Antimicrobial Activity and Protects Against In Vitro and In Vivo Oxidative Stress

**DOI:** 10.3390/foods9010005

**Published:** 2019-12-19

**Authors:** Luisa Pozzo, Rossella Russo, Stefania Frassinetti, Francesco Vizzarri, Július Árvay, Andrea Vornoli, Donato Casamassima, Marisa Palazzo, Clara Maria Della Croce, Vincenzo Longo

**Affiliations:** 1Institute of Agricultural Biology and Biotechnology (IBBA), National Research Council, 56124 Pisa, Italy; rossella.wilbur@gmail.com (R.R.); stefania.frassinetti@ibba.cnr.it (S.F.); vornolia@ramazzini.it (A.V.); clara.dellacroce@ibba.cnr.it (C.M.D.C.); v.longo@ibba.cnr.it (V.L.); 2Department of Agricultural, Environmental and Food Sciences, University of Molise, 86100 Campobasso, Italy; francesco.vizzarri@uniba.it (F.V.); casamassima.d@unimol.it (D.C.); m.palazzo@unimol.it (M.P.); 3Department of Agricultural and Environmental Sciences, University of Bari Aldo Moro, 70126 Bari, Italy; 4Department of Chemistry, Faculty of Biotechnology and Food Sciences, Slovak University of Agriculture, 94976 Nitra, Slovakia; julius.arvay@uniag.sk; 5Cesare Maltoni Cancer Research Center, Ramazzini Institute, Bentivoglio, 40138 Bologna, Italy

**Keywords:** wild Italian *Prunus spinosa* L. fruit, blackthorn, phenolic compounds, antimicrobial, antioxidant

## Abstract

Polyphenol-rich foods could have a pivotal function in the prevention of oxidative stress-based pathologies and antibacterial action. The purpose of this study was to investigate the in vitro antimicrobial activity, as well as the in vitro and In Vivo antioxidant capacities of wild *Prunus spinosa* L. fruit (PSF) from the southeast regions of Italy. The total phenolic content (TPC) was quantified, and the single polyphenols were analyzed by HPLC-DAD, showing high rutin and 4-hydroxybenzoic acid levels, followed by gallic and trans-sinapic acids. PSF extract demonstrated antimicrobial activity against some potentially pathogenic Gram-negative and Gram-positive bacteria. Besides, we investigated the cellular antioxidant activity (CAA) and the hemolysis inhibition of PSF extract on human erythrocytes, evidencing both a good antioxidant power and a marked hemolysis inhibition. Furthermore, an In Vivo experiment with oxidative stress-induced rats treated with a high-fat diet (HFD) and a low dose of streptozotocin (STZ) demonstrated that PSF has a dose-dependent antioxidant capacity both in liver and in brain. In conclusion, the wild Italian *Prunus spinosa* L. fruit could be considered a potentially useful material for both nutraceutical and food industries because of its antioxidant and antimicrobial effects.

## 1. Introduction 

Considering that traditional foods are increasingly believed healthy and wholesome, food manufacturers are developing new food products returning to natural products and traditional recipes that will be attractive to the widest potential consumers [1,2]

Blackthorn (*Prunus spinosa* L.), which belongs to the *Rosaceae* family, is a perennial plant originally growing in temperate continental climate of the northern hemisphere, particularly widespread in the Mediterranean countries and in the southeast regions of Italy. It is used for treatment of many diseases due to its diuretic, spasmolytic, antimicrobial, and antioxidant activities [3]. Moreover, *Prunus spinosa* L. fruit (PSF) is used for the production of various traditional jams and beverages such as juice, wine, tea, and distillates in food industry [4]. It contains substantial quantities of phenolic antioxidants, including, in particular, flavonols, phenolic acids, and coumarin derivatives [5].

Epidemiological investigations demonstrated that diets rich in plant polyphenols protect against diabetes, osteoporosis, cardiovascular, and neurodegenerative diseases [6]. Dietary compounds and specific polyphenol-rich foods could have a pivotal function in the prevention of diseases associated with oxidative stress by increasing the circulation of antioxidant compounds and neutralizing the reactive oxygen species, due to their number and position of hydroxyl groups [7]. Protein nitration, lipid peroxidation, chronic inflammation, and oxidative damage to DNA may be prevented by polyphenols, which results in vasodilatory, vasoprotective, anti-atherogenic, antithrombotic, and anti-apoptotic effects, as free radical scavengers, metal chelators, inhibitors of pro-inflammatory enzymes, and modifiers of cell signaling pathways [8].

Recently, new alternatives have become desirable, and plants metabolites have been screened for antimicrobial agents for treatment of infectious diseases, due to the development of antibiotic resistance by pathogenic bacteria [9]. For instance, in several studies, dietary polyphenols have been reported to exert an antibacterial activity [10].

In the present study, the potential biological activities of wild blackthorn fruit from southeast regions of Italy were investigated. Considering the PSF as a potential natural source of phenolic compounds, this work was designed to study its in vitro antimicrobial, antioxidant, and antihemolytic activities. Moreover, for the first time, the In Vivo protective effect and antioxidant capacity of PSF in high-fat diet (HFD) and streptozotocin (STZ)-induced oxidative stress has been studied. 

## 2. Materials and Methods

Blackthorn fresh fruits were obtained from wild orchards of the Campobasso (Italy) area in October 2015. Taxonomic identification of plant material was confirmed by Prof. Elisabetta Brugiapaglia from Department of Agricultural, Environmental and Food Sciences, University of Molise, Campobasso, Italy. 

### 2.1. Chemicals and Reagents

All solvents and chemicals were of analytical grade. Nutrient Broth (NB), Nutrient Agar (NA), Mueller Hinton Broth (MHB), Mueller Hinton Agar (MHA), McFarland standard were purchased from Oxoid (Basingstone, UK). 6-hydroxy-2,5,7,8-tetramethylchroman-2-carboxylic acid (Trolox), 2,2′-azobis (2-amidinopropane) dihydrochloride (AAPH), 2,7-dichlorodihydrofluorescein diacetate (DCFH-DA), dinitrophenylhydrazine (DNPH), trichloroacetic acid (TCA), perchloric acid (PCA), thiobarbituric acid (TBA), 1,1,3,3-tetramethoxypropane (TEP), guanidine hydrochloride, ortho-phthalaldehyde (OPA), reduced glutathione (GSH), phosphoric acid, potassium dihydrogen phosphate (KH_2_PO_4_), hydrochloric acid (HCl), streptozotocin (STZ), ethanol, ethyl acetate, and methanol from Sigma-Aldrich (St. Louis, MO, USA). All HPLC analytical standards, including protocatechuic acid, syringic acid, rutin, ellagic acid, cynaroside, daidzein, neochlorogenic acid, chlorogenic acid, vitexin, trans *p*-coumaric acid, trans-sinapic acid, trans ferulic acid, rosmarinic acid, resveratrol, apigenin, myricetin, quercetin, and kaempferol, were bought from Sigma-Aldrich (St. Louis, MO, USA). Phosphate buffer saline (PBS) was bought from VWR (Radnor, PA, USA).

### 2.2. Plant Material Preparation 

After the pits were removed, the frozen fruits were lyophilized and crushed in a mortar, and the powder was stored at −20 °C. Briefly, 1 g of sample was extracted with 10 mL of water for 2 h on a horizontal shaker Unimax 2010 (Heidolph Instruments, GmbH, Schwabach, Germany) for in vitro antioxidant activity, polyphenols quantification, and antimicrobial activity. PSF extracts were centrifuged (2300× g at 4 °C for 20 min) (Jouan CR3i centrifuge, Newport Pagnell, UK), and the supernatants were collected. 

For the In Vivo experiment, the lyophilized and powdered fruit was dissolved in water.

### 2.3. Total Phenolic Content (TPC) and Polyphenols Quantification by HPLC-DAD 

For the determination of total phenolic content in water extract, we followed the Singleton and colleagues protocol [11]. The concentration of polyphenols was expressed as mg of gallic acid equivalents (GAE)/g of dry weight (d.w.).

Prior to HPLC analysis, the extracts were filtered through syringe filters Q-Max (0.22 µm, 25 mm, PVDF) (Frisenette ApS, Knebel, Denmark) into the HPLC vials. The HPLC apparatus consisted of an Agilent 1260 Infinity HPLC (Agilent Technologies GmbH, Waldbronn, Germany) quaternary solvent manager coupled with degasser (G1311B), sampler manager (G1329B), Diode Array Detector (G1315C), column manager (G1316A). The analytical column was a Waters Cortecs endcapped RP-C18 column (150 mm × 4.6 mm × 2.7 µm particle size; Waters Corp., Milford, MA, USA). The analyses were carried out at 30 °C by a gradient system with a mobile phase of 0.1% ortho-phosphoric acid in deionised water (C) and acetonitrile gradient grade (D) at a flow rate of 0.60 mL/min, and the injection volume was 5 µL. The gradient elution was as follows: 0–1 min (90% C and 10% D), 1–5 min (85% C and 15% D), 5–10 min. (80% C and 20% D), 10–12 min. (80% C and 20% D), 12–20 min (30% C and 70% D), and 20–25 min (30% C and 70% D). The post-run was set at 3 min. The samples were kept at 4 °C in the sampler manager. The detection wavelengths were set at 265 nm (gallic acid, 4-hydroxibenzoic acid, rutin, and genistein), 320 nm (chlorogenic acid, caffeic acid, trans-*p*-coumaric acid, trans-sinapic acid, trans-ferulic acid, rosmarinic acid, resveratrol), and 372 nm (myricetin, quercetin and kaempferol). Data were analyzed by Agilent Open Lab Chem Station software for LC 3D systems.

### 2.4. HPLC-DAD Method Validation

Reference phenolic compounds were dissolved in HPLC purity methanol and diluted to appropriate concentration ranges (5–50 µg/mL). The linearity of each calibration curve was assessed by linear regression analysis. The limit of detection (LOD) and quantification (LOQ) were estimated by measuring signal-to-noise ratio of the individual peak of each standard compound. The LOD and LOQ were calculated according to the International Conference on Harmonisation guidelines [12].

### 2.5. Antimicrobial Activity

#### 2.5.1. Growth Conditions of Pathogenic Bacteria 

The bacterial strains were supplied by the American Type Culture Collection (ATCC). The antimicrobial activity of PSF extract was studied on three Gram-negative bacteria, specifically *Escherichia coli* (ATCC 25922), *Salmonella enterica* ser. *typhimurium* (ATCC 14028), and *Enterobacter aerogenes* (ATCC 13048), and two Gram-positive bacteria, *Enterococcus faecalis* (ATCC 29212) and *Staphylococcus aureus* (ATCC 25923). 

#### 2.5.2. Antimicrobial Activity

The growth inhibition of selected bacteria exerted by PSF extract was determined according to Delgado Adámez and colleagues [13], with some modifications. 

The tested bacteria were cultured in MHB at 37 °C for 16 h and diluted to match the turbidity of 0.5 McFarland standard. Fifty microliters of bacterial suspensions (about 1–5 × 10^5^ CFU/mL) was added to 100 µL of MHB and to 100 µL of blackthorn extract (0, 0.25, 0.50, 0.75, and 1.00 mg/mL) in a 96-well plate. A negative control was included on each microplate. A positive control of bacterial growth inhibition consisting of two antibiotics, vancomycin (10 µg/mL) for Gram-positive and gentamicin (10 µg/mL) for Gram-negative bacteria was added to the microplate. The plates were incubated at 37 °C for 24 h. Afterwards, the optical density (OD) at 600 nm was determined by a microplate reader (Eti-System fast reader Sorin Biomedica, Modena, Italy). The percentage of growth inhibition was calculated as follows: % growth inhibition = 100 − (OD_S_/OD_C_) × 100(1)
where OD_S_ is the optical density of the sample and OD_C_ is the optical density of the negative control (PSF 0 mg/mL).

### 2.6. In Vitro Antioxidant Activity in Red Blood Cells (CAA-RBC) and Hemolysis Test

According to the regulations of “Fondazione G. Monasterio CNR-Regione Toscana”, human blood samples were obtained from three healthy volunteers in ethylenediaminetetraacetic acid (EDTA)-treated tubes and centrifuged (2300× g at 4 °C for 10 min). Plasma and buffy coat were removed, and erythrocytes were washed twice with PBS pH 7.4.

The antioxidant activity of PSF extract (100 mg/mL) was evaluated in an in vitro system with a modified assay in red blood cells as described by Frassinetti and colleagues [14]. Each value was express as CAA units, as follows [12]: CAA unit = 100 − (∫SA\∫CA) × 100 (2)
where ∫SA is the integrated area of the sample curve and ∫CA is the integrated area of the control curve. 

Hemolysis of PSF extract (100 mg/mL) was analyzed according to the protocol described by Frassinetti and colleagues [15] using AAPH, a generator of peroxyl radicals, to cause the red blood cell lysis. The values reported are the percentage of hemolysis compared with the control.

### 2.7. Animal Study

#### 2.7.1. In Vivo experiment

Male Wistar rats (200–230 g b.w.) were maintained with ad libitum access to food and drinking water for a 12 h light/dark cycle in cages at room temperature with the 55% relative humidity. Rats were divided into two groups: the control (CTR) group (*n* = 5), fed a standard diet (64% carbohydrates, 19% proteins, 7% minerals and vitamins, 6% fibers, and 4% fats; the fats percentage corresponds to the 11% of the diet-derived energy) and the high-fat diet (HFD) group, fed a high fat/cholesterol diet (48.7% carbohydrates, 28% fats, including 2% cholesterol, 13.8% proteins, 4.4% fibers, 5.1% minerals and vitamins, the fats percentage corresponds to the 55% of diet-derived energy). After 5 weeks, the animals of HFD group were treated with a single i.p. injection of streptozotocin (40 mg/kg) [16]. Twenty four rats, resulted to be diabetic with a plasma glucose concentration >250 mg/dL, continued to be fed a HFD diet for a further 4 weeks and were randomly divided into three groups: HFD (n = 8) group, PSF400 (*n* = 8) group, and PSF800 (*n* = 8) group (Figure 1). Rats from CTR and HFD groups were intragastrically administered the same volume of water; rats from PSF400 and PSF800 groups were intragastrically administered lyophilized PSF at different doses (400 mg/kg b.w. and 800 mg/kg b.w., respectively). The weight gain was calculated by initial and final weights. The rats were sacrificed and blood samples were collected by cardiac puncture under general anesthesia. Liver and brain tissues were stored at −80 °C. Hepatic lipids were quantified and oxidative stress markers were analyzed in liver and brain. Blood was centrifuged (2300× g for 15 min) to obtain serum samples for laboratory analysis. Local Ethical Committee approved all animal procedures in accordance with the European Communities Council Directive of 24 November, 1986 (86/609/EEC).

#### 2.7.2. Biochemical Analysis

Serum analyses were performed by a semi-automatic analyzer for clinical chemistry (model ARCO, Biotecnica Instruments SPA, Rome, Italy) for aspartate aminotransferase (AST), alanine aminotransferase (ALT), total cholesterol, and triglycerides. Glucose levels were measured with a glucose meter (Accu-Chek^®^ Roche, Mannheim, Germany), and insulin using a Rat Insulin ELISA kit (Mercodia, Uppsala, Sweden).

#### 2.7.3. Hepatic Lipids Quantification

Fat content of the liver samples was determined by Folch and colleagues protocol [17], slightly modified. Liver samples from rats were homogenized with equal volumes of water and methanol. The resulting homogenate was subjected to three subsequent extractions in chloroform, followed by two washes with KCl 1 M and water. After complete evaporation and prolonged drying of the chloroform, fat content was weighed and expressed as mg/g tissue. 

#### 2.7.4. Oxidative Stress

Malondialdehyde (MDA) concentration of liver and brain samples was analyzed according to Seljeskog and colleagues [18], with some adaptations. An aliquot of 100 µL of homogenate sample was mixed with 0.1125 N PCA (300 μL) and 40 mM TBA (300 μL) for 10 sec and placed in a boiling water bath for 60 min. Methanol (600 μL) and 20% TCA (*w*/*v*) (200 μL) were added to the suspension and mixed for 10 sec, after cooling in a freezer at −20 °C for 20 min. The MDA content was quantified in the supernatant (7000× g for 6 min) by fluorimeter (Perkin Elmer LS-45, Perkin Elmer, Walham, MA, USA) (λ_ex_ = 525, λ_em_ = 560). A standard curve was prepared by dissolving hydrolyzed TEP in water at different concentrations (33.5, 16.8, 8.4, 4.20, 2.10, 1.05, and 0.52 μM). The results have been expressed as nmol MDA/g tissue.

The protein carbonylation was determined using the method adapted from Terevinto and colleagues [19]. Liver and brain samples were homogenized and incubated with 0.02 M DNPH in 2 M HCl. Proteins were then precipitated by adding 20% TCA and recovered by centrifugation (625× g for 10 min). Pellets were washed three times with ethanol:ethyl acetate (1:1, *v*/*v*), melted in 6 M guanidine HCl in 0.02 M KH_2_PO_4_ (pH 6.5), and centrifuged. The absorbance of the supernatant was measured at 390 nm. The results have been expressed as nmol/g tissue.

The GSH content in liver and brain samples was evaluated according to Browne and Armstrong [20], with slight modifications. Proteins were then precipitated by adding 10% TCA (*w*/*v*) at 4 °C for 30 min. An aliquot of 150 µL of the sample was incubated with an equal volume of *o*–phthaldehyde (1 mg/mL) in 10% methanol (*v*/*v*) by 15 min at 37 °C. After centrifugation (625× g for 3 min), fluorescence was measured (Perkin Elmer LS-45, Perkin Elmer, Walham, MA, USA) (λ_ex_ = 350, λ_em_ = 420). A calibration curve has been performed by dissolving GSH in water at different concentrations (50, 25, 12.5, 6.25, 3.13, 1.56, 0.78 µM), and GSH concentrations have been calculated as μmol GSH/g tissue.

### 2.8. Statistical Analysis

The statistical analyses have been performed by Statistical Package for Social Science (SPSS) 17 for Windows (SPSS, Inc., Chicago, IL, USA). The results are shown as the mean value ± standard deviation (s.d.) and analyzed through a one-way ANOVA and Tukey’s test for post-hoc with significance at *p* ≤ 0.05. 

## 3. Results and Discussion

### 3.1. Quantification of Total Polyphenols

The TPC of wild Italian blackthorn fruit extract was quantified by a spectrophotometric method, and the content was 5.50 ± 0.19 mg GAE/g d.w. To our knowledge, any other results have been found about total phenolic content of blackthorn fruit on dry weight, but some authors reported that TPC in blackthorn fruit on fresh weight ranged from 0.42–4.13 mg GAE/g [21,22,23].

The HPLC-DAD method validation was estimated by quantifying the limit of detection (LOD), limit of quantification (LOQ), and the recovery. All parameters indicate that the method exhibits a good sensitivity for identification as well as quantification of the polyphenols. All the parameters are listed in Table 1.

The quality of phenolic profile of wild Italian blackthorn and the concentrations of single compounds are shown in Table 2. Rutin (183.94 mg/kg d.w.) was the principal phenolic component, followed by 4-hydroxybenzoic acid, gallic acid, trans-sinapic acid, quercetin, trans-ferulic acid, caffeic acid, rosmarinic acid, trans cumaric acid, genistin, and myricetin. Our findings are partially in accordance with those of some other authors that showed considerable quantities of phenolic acids (quercetin and caffeic acid) in blackthorn fruits from Southeast Serbia [3,24]; by contrast we did not find either neochlorogenic or kaempferol. HPLC-UV analysis of the methanolic extract of fresh blackthorn plums from Turkey recently allowed Baltas and colleagues to detect five phenolic acids, namely protocatechuic acid, p-OH benzoic acid, vanillic acid, syringic acid, and p-coumaric acid, as well as flavonoids, such as epicatechin and luteolin [25]. Another recent study about quantification of phenolic compounds by HPLC-UV in methanolic extract of frozen blackthorn fruits from Romania showed high chlorogenic and neochlorogenic acid levels, followed by glycosides of quercetin [23]. Considering that the solubility of polyphenols in solvent of different polarity is determined by their structure, different types of extraction solvent and procedures may influence the efficiency of phenolic compounds extraction and their resultant content [3].

### 3.2. Antimicrobial Activity

The antimicrobial activity on selected Gram-negative (Figure 2A) and Gram-positive (Figure 2B) enteric bacteria was measured by evaluating the growth inhibition by increasing concentrations of PSF extract. The antimicrobial activities have been compared with the standard antibiotics, used as positive controls. 

The lowest concentration of tested PSF extract (0.25 mg/mL) inhibited more than 50% of the Gram-negative bacteria *Escherichia coli* (70.19% ± 1.21%), *Salmonella typhimurium* (79.98% ± 0.54%), and *Enterobacter aerogenes* (83.02% ± 0.54%) growth (Figure 2A). The same concentration (0.25 mg/mL) was able to inhibit more than 50% of the Gram-positive bacteria *Enterococcus faecalis* (82.86% ± 1.94%) and *Staphylococcus aureus* (79.92% ± 1.23%) growth (Figure 2B). The antimicrobial activity of phenolic compounds occurring in plant foods has been widely studied against a wide range of microorganisms. The damage to the bacterial membrane and suppression of some virulence factors, including enzymes and toxins, are suggested to be the mechanisms of their antimicrobial action [26]. Some flavonoids (rutin, myricetin, and quercetin) and phenolic acids (gallic, caffeic, and ferulic acids) of PSF extract may be responsible for its antibacterial action [27,28].

### 3.3. In Vitro Antioxidant Activity

As shown in Figure 3A, pretreated erythrocytes with PSF aqueous extract (100 mg/mL) exhibited a significantly higher cellular antioxidant activity (CAA unit = 48.43 ± 1.68) compared with untreated cells (CAA = 0; *p* ≤ 0.001), comparable to 100 µM Trolox (CAA unit = 16.52 ± 3.60; *p* ≤ 0.001) and 500 µM Trolox (CAA unit = 36.67 ± 1.48; *p* ≤ 0.001). Taking these results into consideration, the EC50 of PSF extract for antioxidant activity in red blood cells was 100 mg/mL.

The antihemolytic activity of PSF extract was screened in erythrocytes exposed to high doses of AAPH, causing a strong oxidative hemolysis. Figure 3B shows that PSF extract exerted a significant inhibition of AAPH-induced hemolysis compared with the control erythrocytes (AAPH-treated). PSF extract (100 mg/mL) pretreated cells demonstrated a marked antihemolytic effect (84% hemolysis inhibition) compared with AAPH-treated cells (*p* ≤ 0.001), with a reduction of the hemolysis similar to that of the highest concentration of the reference standard (500 µM Trolox). The antihemolytic EC50 of PSF extract was 10 mg/mL (data not shown). We found that PSF exerted a potent ROS-scavenger activity. Indeed, when intact human erythrocytes were pre-incubated with a PSF aqueous extract, a strong protective effect against AAPH-generated ROS production and hemolysis was observed. These antioxidant and antihemolytic effects of PSF are probably due to the activity of gallic acid, rutin, and quercetin in red blood cell [29,30].

### 3.4. In Vivo Experiment

#### 3.4.1. The Effect of Blackthorn on Body Weight and Liver Weight

In comparison with CTR group, rats of the HFD group exhibited a significant lower final body weight (396.8 ± 40.6 vs. 307.5 ± 23.3 g/rat, respectively) (*p* ≤ 0.001). The administration of PSF did not induce significant changes in the final body weight, neither in PSF400 group (317.2 ± 27.6 g/rat), nor in PSF800 group (312.7 ± 42.5 g/rat), when compared with HFD group.

However, when compared with CTR rats, HFD rats exhibited a statistically significant increase in liver weight (8.9 ± 1.4 vs. 13.4 ± 1.2 g, respectively) (*p* ≤ 0.001) and in relative liver weight (2.2 ± 0.2 vs. 4.1 ± 1.0 g liver/100 g b.w., respectively) (*p* ≤ 0.001). No significant difference in liver weight was found between HFD-fed rats and PSF-treated rats of PSF400 group (14.8 ± 1.7 g) and PSF800 group (14.6 ± 1.8 g). The same trend was found in relative liver weight between HFD-fed rats and PSF-treated rats of PSF400 group (4.5 ± 0.3 g liver/100 g b.w.) and PSF800 group (4.8 ± 0.7 g liver/100 g b.w.). HFD treatment caused hepatic lipid accumulation and increased liver weight and all the biochemical parameters in serum [16,31]. However, the PSF extract did not improve the liver weight and serum and liver biochemical parameters linked to steatosis. 

#### 3.4.2. The Effect of PSF on Serum and Liver Biochemical Parameters

Serum AST, ALT, glucose, total cholesterol, triglycerides, and total hepatic lipid content were significantly higher in the HFD group compared with the normal diet group (CTR group), while serum insulin was significantly lower. After four weeks of treatment with 800 mg of PSF/kg b.w., rats of the PSF800 group showed a significant decrease of total hepatic lipids content compared with HFD group (162.15 ± 35.52 vs. 209.90 ± 11.91; *p* ≤ 0.05) (Table 3). Some studies have demonstrated that polyphenols decrease the hepatic lipid accumulation caused by high-fat diet [32]. Moreover, it was also reported that the single isolated polyphenol can improve the high liver lipids content due to a high-fat diet administration, as in the case of rutin [33], gallic acid [34], and quercetin [35]. The crude extracts can be more advantageous than the isolated components, since a single bioactive molecule can change its properties with the presence of other compounds in the extracts [36].

#### 3.4.3. The Effect of PSF on Liver and Brain Oxidative Stress of Rats

The high-fat diet was probably responsible for the decrease of GSH (Figure 4A) content and the increase of protein carbonylation (Figure 4C) and MDA (Figure 4E) levels in the liver samples of HFD rats, compared with the CTR group. Moreover, while hepatic GSH content was not affected by PSF treatment (Figure 4A), administration of PSF improved the oxidative stress status of rats according to the protein carbonylation, at the higher concentration of treatment (800 mg/kg b.w.) (Figure 4C), and to the MDA content, in a dose-dependent manner (Figure 4E). It has been demonstrated that plant polyphenols are related to the improvement of hepatic oxidative stress caused by a high-fat diet through the e activation of Nrf2 transcription factor, which increases expression of antioxidant enzymes [37]. Moreover, it was reported that even the single polyphenol, if isolated, can improve the high-fat-diet-induced hepatic oxidative stress, as in the case of rutin [33] and gallic acid [38].

In comparison with the CTR group, HFD treatment promoted an increase of the brain oxidative stress parameters in rats, as shown by protein carbonylation (Figure 4D) and MDA assay (Figure 4F). The addition of 400 mg/kg b.w. and 800 mg/kg b.w. of PSF to the diet reversed the effect caused by the high-fat diet and, in particular, the MDA assay showed a dose-dependent response pattern. Nevertheless, both the HFD and the PSF treatments did not induce significant changes in rat brain GSH levels (Figure 4B). The intake of a high-fat diet is linked to an increased risk of neurodegenerative disease related to diabetes [39]. Considering that, the polyphenols-rich fruits could protect neurons against the oxidative stress induced by intake of saturated fatty acids [40]. Recently, Nabavi and colleagues demonstrated that gallic acid exerts a neuroprotective effect against sodium fluoride-induced oxidative stress in rat brain [41]. Moreover, it has been shown that other polyphenols contained in PSF, such as rutin, ferulic acid, and trans-sinapic acid, should contribute to the prevention of brain oxidative stress in rats [42,43,44].

Our findings suggest an improved liver and brain antioxidant defense in rats treated with PSF.

## 4. Conclusions

All in all, our findings indicated that wild Italian blackthorn fruit is rich in polyphenol compounds, shows an in vitro antioxidant activity, and exhibits a selective growth inhibition of some potentially pathogenic bacteria strains. Moreover, this study is the first to evaluate an In Vivo antioxidant activity of PSF. In particular, our findings indicated that the oxidative stress arising in HFD group is decreased in liver and brain tissues by the intake of blackthorn fruit. The PSF supplementation demonstrated In Vivo antioxidant capacities, reducing liver and brain oxidative stress, probably due to the presence of polyphenols, such as rutin, 4-hydroxybenzoic acid, gallic acid, trans-sinapic acid, quercetin, trans-ferulic acid, caffeic acid, rosmarinic acid, trans coumaric acid, genistin, and myricetin, which were identified in the blackthorn fruit.

Thus, it is supposed that the regular consumption of wild Italian blackthorn fruit should increase the circulation of bioactive compounds, such as polyphenols, which could possibly improve the endogenous antioxidant system and protect tissues against oxidative stress damage induced by high-fat diet and hyperglycemia. Considering its beneficial properties, wild Italian blackthorn fruit can be potentially used to produce natural functional food, novel nutraceuticals, and it can also be employed in food processing.

## Figures and Tables

**Figure 1 foods-09-00005-f001:**
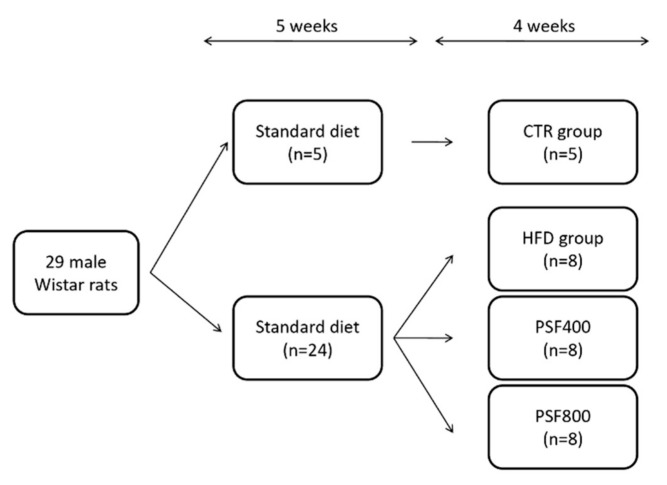
Study design of the In Vivo experiment. CTR, control group; HFD, high-fat diet group; PSF400, PSF800.

**Figure 2 foods-09-00005-f002:**
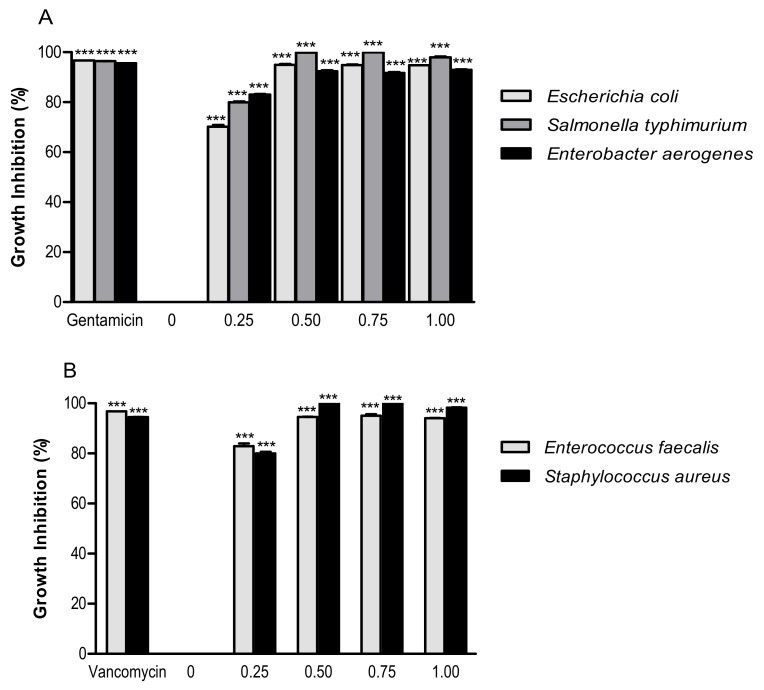
Growth inhibition effect of PSF extract (0, 0.25, 0.50, 0.75, and 1.00 mg/mL) against Gram-negative bacteria (**A**) (*Escherichia coli* ATCC 25922, *Salmonella enterica* ser. *typhimurium* ATCC 14028, and *Enterobacter aerogenes* ATCC 13048) and Gram-positive bacteria (**B**) (*Enterococcus faecalis* ATCC 29212 and *Staphylococcus aureus* ATCC 25923). Significantly different from negative control (PSF 0 mg/mL): *** *p* ≤ 0.001. Results are reported as means (*n* = 3) values ± standard deviation.

**Figure 3 foods-09-00005-f003:**
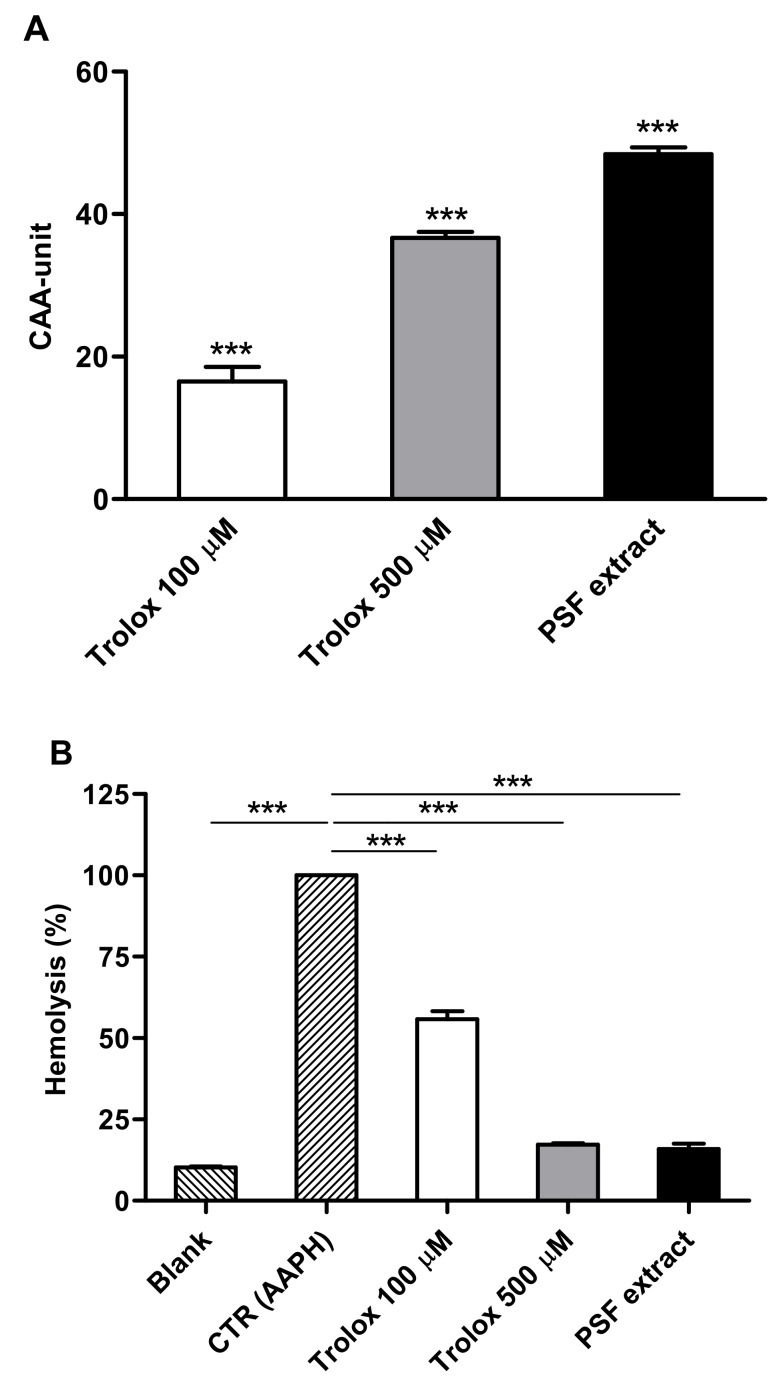
(**A**) Effects of PSF extract (100 mg/mL) on cellular antioxidant activity (CAA) in human erythrocytes. Significantly different from untreated cells (CAA unit = 0): *** *p* ≤ 0.001. (**B**) Effects of PSF extract (100 mg/mL) on dihydrochloride (AAPH)-induced oxidative hemolysis in human erythrocytes. Significantly different from CTR (AAPH-treated cells): *** *p* ≤ 0.001. Trolox was used as reference standard. Results are reported as means (*n* = 3) values ± standard deviation.

**Figure 4 foods-09-00005-f004:**
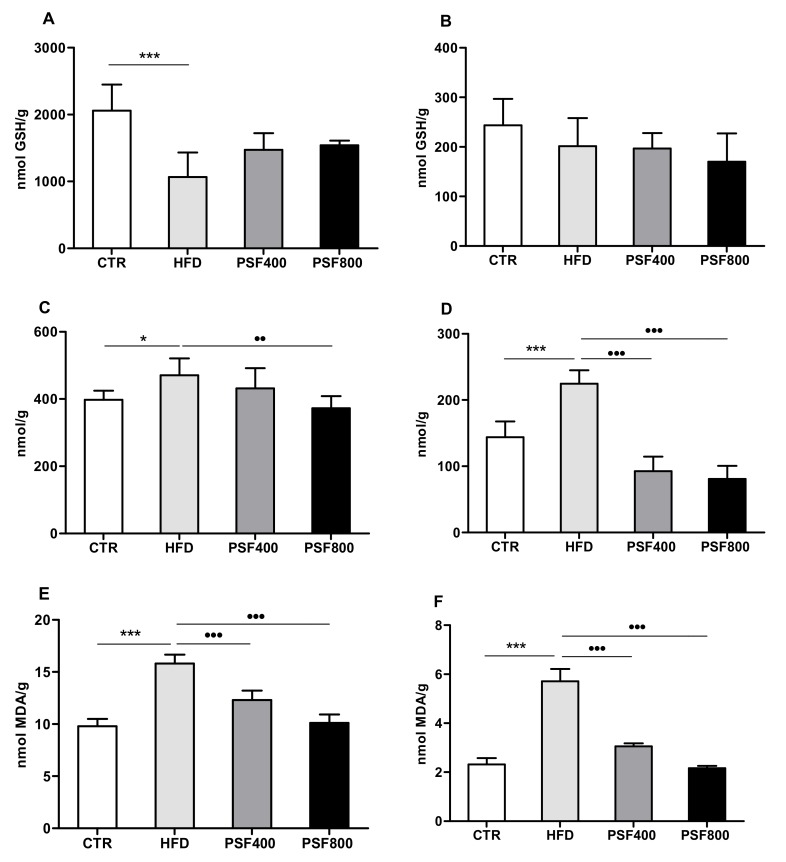
Effect of PSF treatment at two different concentrations (PSF400 and PSF800) on GSH content of liver (**A**) and brain (**B**), protein carbonylation of liver (**C**) and brain (**D**), and malondyaldeide of liver (**E**) and brain (**F**). Results are reported as means (*n* = 7) values ± standard deviation. Values within each row of different letters are significantly different (*p* ≤ 0.05), *p* ≤ 0.05 vs. CTR; **, *p* ≤ 0.001 vs. CTR; ^●●^, *p* ≤ 0.01 vs. HFD; ^●●●^, *p* ≤ 0.001 vs. HFD.

**Table 1 foods-09-00005-t001:** Retention time (Rt), LOD (limit of detection), LOQ (limit of quantification), and recovery of phenolic compound quantification method by HPLC-DAD in the *Prunus spinosa* L. fruit (PSF) aqueous extract (*n* = 3).

Phenolic Compound	Rt (min)	LOD (ug/mL)	LOQ (ug/mL)	Recovery (%)
Gallic acid	2.860	0.012	0.033	98.2 ± 0.81
Rutin	5.909	0.009	0.030	89.1 ± 0.89
4-hydroxibenzoic acid	7.112	0.005	0.017	101.2 ± 1.01
Caffeic acid	8.361	0.008	0.027	97.5 ± 0.99
Trans p-coumaric acid	11.741	0.004	0.013	96.0 ± 0.80
Trans -ferulic acid	12.981	0.003	0.010	97.8 ± 1.08
Trans-sinapic acid	13.062	0.011	0.037	98.2 ± 1.15
Myricetin	17.081	0.015	0.050	99.5 ± 0.88
Rosmarinic acid	17.463	0.009	0.023	102.1 ± 0.96
Quercetin	18.853	0.090	0.299	99.5 ± 0.95
Genistein	19.811	0.009	0.031	91.5 ± 0.77

**Table 2 foods-09-00005-t002:** Concentrations of phenolic compounds in the PSF aqueous extract.

Phenolic Compound	Concentration (mg/kg d.w.)
Gallic acid	41.10 ± 3.68
Rutin	183.94 ± 0.45
4-hydroxybenzoic acid	73.93 ± 0.06
Caffeic acid	3.36 ± 0.36
Trans *p*-coumaric acid	2.99 ± 0.02
Trans-ferulic acid	4.93 ± 0.07
Trans-sinapic acid	37.69 ± 0.05
Myricetin	1.47 ± 0.03
Rosmarinic acid	3.23 ± 0.03
Quercetin	9.94 ± 0.01
Genistin	1.74 ± 0.00

**Table 3 foods-09-00005-t003:** Nutritional effect of PSF on biochemical parameters in serum and liver of rats (*n* = 7).

	CTR	HFD	PSF400	PSF800
AST (UI/dl)	93.98 ± 7.04	194.33 ** ± 42.90	181.00 ** ± 22.23	168.50 * ± 44.81
ALT (UI/dl)	39.06 ± 10.09	143.17 ** ± 35.15	146.52 ** ± 51.96	133.58 ** ± 36.90
Insulin (µg/l)	1.44 ± 0.80	0.19 ** ± 0.15	0.22 ** ± 0.15	0.20 ** ± 0.03
Glucose (mg/dl)	145.20 ± 20.80	439.67 *** ± 70.21	432.57 *** ± 33.57	443.60 *** ± 43.32
Total cholesterol (mg/dl)	109.30 ± 21.40	236.20 *** ± 45.09	228.36 *** ± 29.13	219.18 *** ± 17.94
Triglycerides (mg/dl)	75.73 ± 7.38	179.80 ** ± 59.30	164.33 ** ± 30.52	170.25 ** ± 21.80
Total hepatic lipids (mg/g)	65.36 ± 9.14	209.90 *** ± 11.91	198.29 *** ± 34.52	162.15 *** § ± 35.52

Analyses were performed through one-way ANOVA and Tukey’s test for post-hoc. *, *p* ≤ 0.05 vs. CTR; **, *p* ≤ 0.01 vs. CTR; ***, *p* ≤ 0.001 vs. CTR; §, *p* ≤ 0.05 vs. HFD. AST, aspartate aminotransferase; ALT, alanine aminotransferase.

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
