# Peer review of "Wild Italian *Prunus spinosa* L. Fruit Exerts In Vitro Antimicrobial Activity and Protects Against In Vitro and In Vivo Oxidative Stress"

_foods, 2019, doi:10.3390/foods9010005_

Round 1

Reviewer 1 Report

This paper is sufficient to be accepted by Foods in that all details in it were updated based on the comments from previous reviewers.

Author Response

Dear Referee,

Thank you for your interesting and useful review.

Reviewer 2 Report

The study of antioxidants capacity does not bring any novelty in the field of bioactive compounds.The study of antimicrobial activity is not enough,as it does not contemplate other pathogens of wide distribution in food processing plants and responsible for serious food infections.

Author Response

Dear Referee,

Thank you for your interesting and useful reviews.

Shown below a list of our response to the Referee’ comments.

Reviewer 3 Report

Dear Authors.

The manuscript is written very well and provides an original scientific work and very interesting results with discussions supported by enough references.

The manuscript need revision for the following:

Line 18: please spell out TPC.

Lines 36-38: Revise to “Blackthorn (Prunus spinosa L.), which belongs to the Rosaceae family, is a perennial … of Italy”

Line 57: from here to the end, it’s better to use ‘blackthorn’ than the Latin name ‘Prunus spinosa’ in the entire manuscript

Lines 83-84: How did you make fruit powder? Did you dry the fruit before lyophilization? This needs more detail description. One g of sample, dried sample ?

Lines 289-290: I believe the sentence is the part of figure 3 caption.

Lines 293-303: Need to include discussion related to the results.

Lines 356-357: “in vivo activity of PSF” on what??

Please check again for grammatical errors in the entire manuscript.

Author Response

Dear Referee,

Thank you for your interesting and useful review.

Shown below a list of our response to your comments.

Round 2

Reviewer 2 Report

The conclusions do not collect whit sufficient extension the results obtained. They should be improved.

This manuscript is a resubmission of an earlier submission. The following is a list of the peer review reports and author responses from that submission.

Round 1

Reviewer 1 Report

This is a straightforward- and clearly-written paper ‘Wild Italian Prunus spinosa L. fruit exerts in vitro antimicrobial activity and protects against in vitro and in vivo oxdative stress’. Results part, I have no major criticism about this manuscript. However, it needs to be improved in a few points .

<Major comments>

Are effects of PSF from the combinatorial effects of all components of PSF? Have you checked the effects of any major components of PSF? Have you compared PSF with with its conpounds like Rutin, etc?

<Minor comments>

Check: Font size First figure legend in page 2? Is it necessary? What is the meaning of the sentence in line 35? What is the meaning of ‘to return to the natural and traditional products’? Can you clarify the meaning and supporting data like market share etc? Please add the full name of chemicals of English. If possible, Table or list of abbreviation. In line 175, please describe the method briefly though the method is referred to the citation.

Question:

Where are the main figures?

Author Response

Please find attached a PFD file

Reviewer 2 Report

In the manuscript entitled » Wild Italian Prunus spinosa L. fruit exerts in vitro antimicrobial activity and protects against in vitro and in vivo oxidative stress « of authors Luisa Pozzo et al., authors described a good piece of work where they studied antimicrobial, antioxidative effects in vitro as in vivo along with phytochemical composition. The design of the experiment, representative number of samples and appropriate evaluation of obtained results were realized.

In my Opinion, the paper is well written, however, I request to modify the absract as there is no main results described, the most important results must be shown including the main phytochemicals.

In page 2; there is no legends in the figures, also for poliphenolic is mistakenly spelled.

Furthermore, the authors reports the amount of substances as mg/kg d.w. So Method validation such, LOD, LOQ, Recovery, Precision, accuracy and uncertainty evaluation of applied method have to be reported in a separate table.

My last question which is about anthocyanins and quinates present in this fruit as described by Varga E [Polyphenolic compounds analysis and antioxidant activity in fruits of Prunus spinosa L.] Acta Pharm Hung 2017;87(1):19-25. So discussion must be improved taking into accounts these results.

Author Response

Please find attached a PDF file 
